# The Prognostic Value of High-Sensitive Troponin T Rise Within the Upper Reference Limit in Breast Cancer: A Prospective Pilot Study

**DOI:** 10.3390/cancers17142412

**Published:** 2025-07-21

**Authors:** Sergey Kozhukhov, Nataliia Dovganych

**Affiliations:** 1SI NSC The M.D. Strazhesko Institute of Cardiology, Clinical and Regenerative Medicine of the National Academy of Medical Sciences of Ukraine, 03680 Kyiv, Ukraine; dovganychnat@gmail.com; 2Cardio-Oncology Center, 03680 Kyiv, Ukraine

**Keywords:** breast cancer, cardiotoxicity, echocardiography, high-sensitivity troponin T

## Abstract

Anti-cancer therapy is often accompanied by adverse reactions, among which cardiovascular toxicity is the most significant. Therefore, it is very important to diagnose cardiotoxicity before myocardial systolic function decreases or symptoms of heart failure appear. In this prospective study of 60 breast cancer patients, we studied dynamic changes in high-sensitive cardiac troponin T below the upper limit of normal. We found that an increase in troponin T > 81% was determined as the optimal threshold value for detecting early biochemical cardiotoxicity. Those patients should be considered as high-risk patients for cardiotoxicity and need more precise cardiac monitoring and early preventive strategies.

## 1. Introduction

Breast cancer (BC) is the most common female cancer worldwide, with an estimated incidence of cases per year in Ukraine [1].

Over the last decade, progress in BC therapy improved short- and long-term survival, [2] but increased cancer therapy-related cardiac dysfunction (CTRCD) [3].

Routine cardiovascular (CV) tests such as electrocardiography and transthoracic echocardiography (TTE) have limited sensitivity and specificity for the early detection of myocardial damage. CV imaging generally detects CTRCD at advanced stages, whereas biomarkers are capable of detecting even minimal cardiomyocyte injury [4]. The most important markers of cardiac damage are cardiac troponins: (cTn) I and cTnT [5]. The upper reference limit of highly sensitive troponin T (hsTnT) values is a specific marker of myocardial damage [6].

Cancer therapy increases the risk of cardiac/myocardial dysfunction and heart failure (HF); therefore, the early detection of CTRCD in cancer patients is necessary to prevent future CV complications [7,8]. ESC guidelines on cardio-oncology recommend cardiac biomarkers assessment (natriuretic peptides and/or cTn) before potentially cardiotoxic therapies, with the class of recommendation I and level of evidence C [9].

Currently, there are no studies demonstrating the clinical significance of increasing hsTnT below the upper limit of normal (ULN), and more research is required.

Therefore, in this study, we investigated CTRCD in patients with BC and hsTnT cut-off < 14 ng/L. We hypothesized that the rise in hsTnT even within ULN has important clinical significance for CV outcomes.

## 2. Material and Methods

### 2.1. Study Design and Participants

This single-center, prospective, observational pilot study recruited patients from January 2018–January 2022 at National Cancer Institute and Cardio-Oncology Center of NSC “The M.D. Strazhesko Institute of Cardiology, Clinical and Regenerative Medicine of the National Academy of Medical Sciences of Ukraine” (Kyiv, Ukraine). Inclusion criteria were newly diagnosed I–III-stage BC, and chemotherapy in neoadjuvant/adjuvant regimens as a part of a complex cancer treatment according to current guidelines. Exclusion criteria were a history and presence of HF and/or left ventricular (LV) dysfunction, LV ejection fraction (LV EF) < 50% of any etiology, atrial fibrillation or flutter, myocardial infarction, stroke, and previous treatment of BC.

All patients signed an informed consent before enrollment into the study. The study protocol was approved by the local ethical committee.

All patients underwent clinical examination, 12-lead ECG, TTE, and hsTnT at baseline, at 3 and 6 months.

### 2.2. Echocardiographic Data Acquisition

TTE was performed using an ultrasound machine (Toshiba Aplio 500, Toshiba, Japan). LV EF was calculated using the modified Simpson method [10]. An LVEF decrease of ≥10% (percentage points) or below its limit value (≤50%), according to recommendations, was considered as a CTRCD [9].

### 2.3. Blood Sampling and Analysis

Blood from all participants was collected in serum tubes, centrifuged, and stored frozen at baseline, in terms of 3 and 6 months. The measurement of hsTnT in plasma samples was performed using the Elecsys^®^ Troponin T hsTnT assay (Roche Diagnostics) on a Cobas^®^ e411 immunoassay analyzer. The limit of blank and 99th percentile cutoff values for the hsTnT assay were 3 and 14 ng/L [11]. Laboratory analyses were performed blinded to clinical status.

### 2.4. Statistical Analysis

All statistical analyses, including correlation and nonlinear regression, ROC curves, and AUC analysis were calculated by the use of SPSS 19.0 (SPSS Inc., Chicago, IL, USA). Results are presented as (M ± m). Data were compared using the t-test. Categorical data are expressed as numbers (%) of subjects. Pearson’s correlation analysis was performed with parametric and non-parametric methods to determine associative relationships between indicators. All the tests were two-sided, and a *p*-value less than 0.05 was considered statistically significant.

## 3. Results

### 3.1. Study Population and Baseline Characteristics

After screening 88 patients, 60 were finally included and analyzed (Figure 1). Excluded patients were managed in accordance with current recommendations.

The mean age of BC patients was 48.6 ± 1.3 years. Patients’ demographic characteristics, comorbidity, and type of cancer therapy are detailed in Table 1 and Table 2.

BC was left-sided in 48.3% of the patients, and the cumulative antracycline dose was 258 ± 13.2 mg/m^2^. Over 6 months, BC patients received from 2 to 7 cycles of HER2 inhibitors according to their hormone status and treatment protocol. In total, 32.0% of the patients received radiotherapy with a radiation dose of 45.7 ± 0.9 Gy. According to ESC guidelines [9], the HFA-ICOS risk assessment was calculated; 7% of patients were classified as moderate, 4% as high, and 89% as low risk for developing cardiotoxicity.

### 3.2. Cardiac hsTnT

Among 88 BC patients, 21 with hsTnT > 14 ng/L were excluded: 4 patients before the start of BC treatment and 17 patients during the follow-up (FU). Finally, we have analyzed 60 patients. The baseline level of hsTnT was 5.5 ± 1.4 ng/L. We calculated the rise in hsTnT (ΔhsTnT) by the difference (%) between its baseline level and during FU—6 months of cancer treatment.

During FU, hsTnT increased in all patients up to 10%–305% from baseline, by an average of 94.2%. The distribution of hsTnT changes during 6 months is shown in Figure 2.

### 3.3. Myocardial Function

LV EF was normal at baseline (61.9 ± 3.3%) in all BC women (Figure 3). In 6 months of cancer treatment, LV EF decreased significantly (56.3 ± 7.0%) in comparison to its baseline value (*p* < 0.045). Moreover, 9 patients (15%) demonstrated an LVEF decrease ≥ 10% (percentage points); among them, 6 patients had an EF below its limit value (≤50%), and EF was lower than 40% in 2 patients.

Based on the analysis of the selected patients with an increase in hsTnT within ULN, in 15% patients LV EF decreased as a criterion for cardiotoxicity. Therefore, we hypothesized that an elevated troponin level even within the ULN reflects cardiomyocyte damage, as evidenced by a decreased systolic function or HF symptoms.

Logistic regression analysis was used to establish the significance of the proposed method for determining early biochemical cardiotoxicity, namely the calculation of a threshold value of Δ hsTnT.

Receiver operating characteristic (ROC) curves were used to assess the diagnostic accuracy for the detection of cardiotoxicity according to the changes in hsTnT and LV EF values during 6 months of cancer treatment. Optimal cut-off values were derived from ROC curves, and sensitivity and specificity values were calculated. The model was also adapted to solve the situation when the sensitivity of the test is greater than its specificity. A drop in LV EF by ≥10% from its baseline level or ≤50% was used as a criterion for CTRCD. The results are presented in Figure 4.

In the ROC curve, the cut-off value for the rise in hsTnT ≥ 81% to indicate the drop in LV EF ≥ 10% with a sensitivity of 89% and specificity of 43% (area under ROC curve = 0.78, 95% CI (0.67–38 0.90), *p* < 0.05. Thus, an increase in the hsTnT concentration during 6 months of cancer therapy is a reliable predictor of myocardial damage, followed by a decline in heart function. The rise in hsTnT ≥ 81% was determined as the optimal threshold value for the detection of early biochemical cardiotoxicity.

Additionally, a logistic regression analysis using the criterion of EF decrease from ≥5 to <10 percentage points was performed. Such an LV EF drop is in a gray zone, and current guidelines do not recommend the use of preventive or therapeutic strategies in those cases. In our opinion, it is important to identify cardiotoxicity early, namely when LV EF decreases by less than <10% (Figure 5).

In those patients, the AUC was 0.84 (*p* = 0.01), 95% CI (0.73–0.95), and hsTnT has a good predictive value for a drop of LV EF.

## 4. Discussion

The primary aim of this prospective pilot study was to determine a new approach that might be able to stratify cancer patients according to the increase in hsTnT within the UNL. The ESC 2022 Cardio-Oncology Guidelines recommend serial biomarkers (an increase in Tn above the ULN) and TTE monitoring of LV EF (decrease in LV EF ≥ 10% or ≤50%) as the best available methods for detecting changes in heart function, which leads to further therapeutic decisions [9]. Patients with CTRCD may already have structural heart damage, and a decrease in systolic function, which can lead to irreversible loss of cardiomyocytes [12,13]. As a result, in some cases, cancer treatments are being withdrawn or withheld. A one-month delay of cancer treatment is associated with increased mortality across surgical, chemotherapy, and RT domains [14].

According to Clerico et al., an increase of around 3–5 ng/L in hsTn concentration correlated to necrosis of about 10–20 mg of myocardial tissue, which cannot be detected with such precision by standard 2-D TTE techniques [15].

Therefore, there is a search for new cardiotoxicity criteria, earlier than the currently accepted ones, which can be used for the stratification of high-risk patients and allow for early preventive medical interventions, cancer treatment modification, or the need for close cardiac monitoring [16,17].

Our study demonstrated that hsTnT levels increased during six months of BC treatment in all patients. We hypothesized that even an increase in hsTnT within ULN during chemotherapy displays myocardial damage and an increase of hsTnT > 81% from the initial value was determined as the optimal threshold value for detecting early biochemical cardiotoxicity. According to the literature, no other study investigated the significance of changes in Tn below ULN in patients with cancer.

Several studies presented different lower Tn cut-off values. In a large study of cancer patients, hsTnT was measured before chemotherapy started, and increased mortality was predictable using a relatively low cutoff of 7 ng/L [18]. A recent larger study found that elevated hsTnT levels of >14 ng/L at anthracycline completion were associated with a 2-fold increased CTRCD risk (hazard ratio, 2.01; 95% CI, 1.00–4.06) [19]. In a study by Skyttä et al., thoracic RT for BC in 58 patients increased Tn levels measured during and at the end of RT by >30% in 21% [20]. A meta-analysis of 61 clinical studies from Michel et al. concluded that Tn elevation is associated with cancer treatment (OR 14.3; CI 6.0–34.1), especially in the case of HER2-therapy, elevated Tn was associated with a high risk of systolic dysfunction (OR 11.9; CI 4.4–32.1). However, Tn cut-off values varied across studies [21].

Another critical clinical question is whether decreasing LV EF from ≥5 to <10 percentage points has prognostic significance and can be a new red flag for future CV complications, especially for LV dysfunction. At least in our study, we observed high sensitivity values at the identified cut-off of hsTnT for such an EF drop criterion.

The criterion of reducing LVEF by more than 5 percentage points, to below 10 percentage points, has been attempted previously, by using cardiac MRI [22]. To date, this criterion has not been validated.

Although this hypothesis remains to be tested in clinical trials, it illustrates the importance of understanding the pre-early biochemical cardiotoxicity by kinetics of circulating hsTnT concentrations with 99th percentile cut-off values of 14 ng/L.

*Study limitations.* Our pilot study was limited by a small sample size, one type of cancer site, and using criteria outside of current guideline recommendations. Patients were evaluated in the context of cardiotoxicity. We did not conduct global longitudinal strain, a long-term follow-up of patients, and had not studied clinical outcomes such as CV death and all causes of death.

## 5. Conclusions

A significant number of BC patients receiving cardiotoxic cancer treatment have objective evidence of myocardial damage or LV myocardial dysfunction. Today, cardiac Tn can be used for the early identification of patients at risk for CTRCD. However, the Tn cut-offs as criteria of cardiotoxicity need to be more clearly established in cancer patients.

We found that hsTnT rise within ULN can be valuable for risk stratification during cancer treatment. BC patients with increased hsTnT plasma concentration > 81% from the baseline value should be considered for more detailed cardiac monitoring and early preventive pharmacological strategies.

Further prospective, randomized, multicenter trials are needed to confirm our findings in BC and other cancer sites.

## Figures and Tables

**Figure 1 cancers-17-02412-f001:**
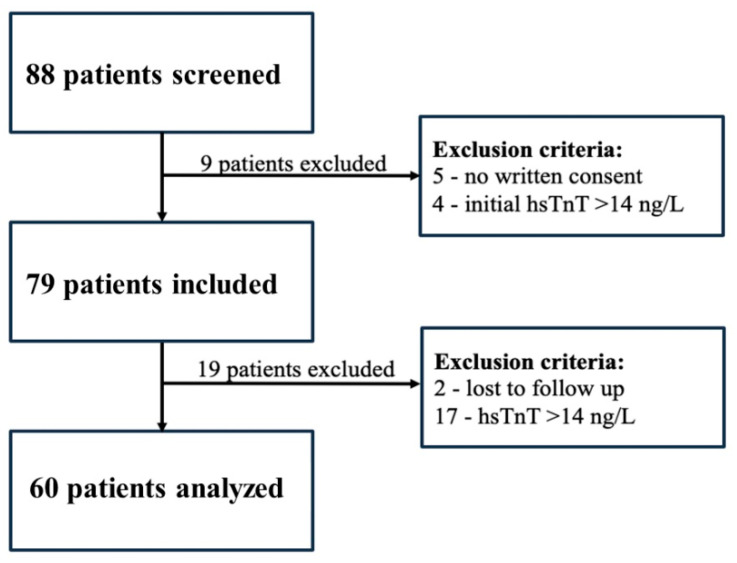
The screening, inclusion, and follow-up of patients. A total of 88 patients were screened, and the main reasons for exclusion were missing written consent (*n* = 5) and baseline hsTnT > 14 ng/L (*n* = 4), while the reasons for loss to follow-up were personal reasons (*n* = 2) and hsTnT > 14 ng/L (*n* = 17). Patients (*n* = 60) were analyzed when at least a baseline and one follow-up hsTnT test was performed.

**Figure 2 cancers-17-02412-f002:**
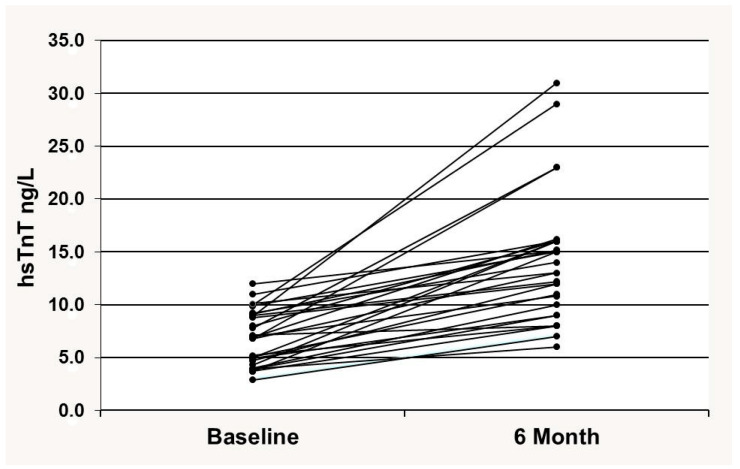
The distribution of hsTnT changes during 6 months of cancer treatment. hsTnT, high-sensitivity troponin T.

**Figure 3 cancers-17-02412-f003:**
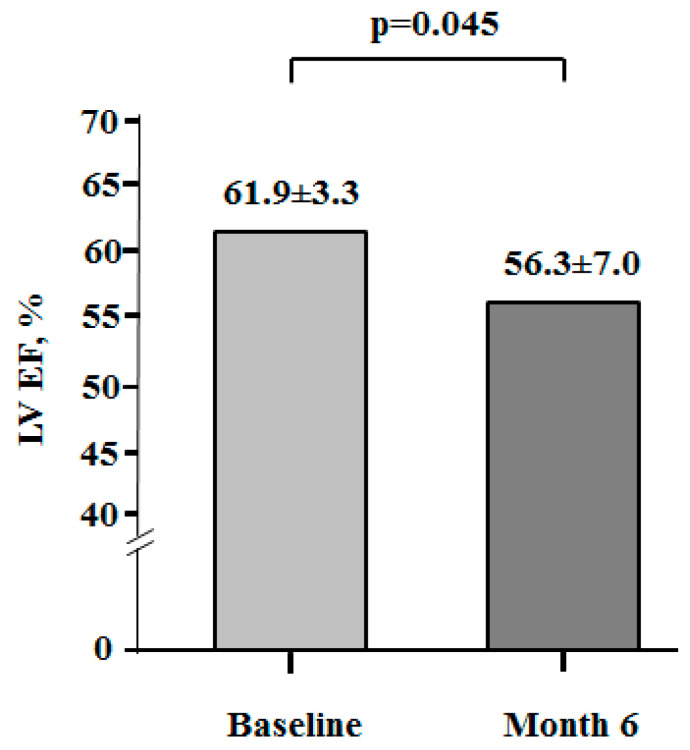
Distribution of LV EF at baseline and month 6. LV EF: Left ventricular ejection fraction.

**Figure 4 cancers-17-02412-f004:**
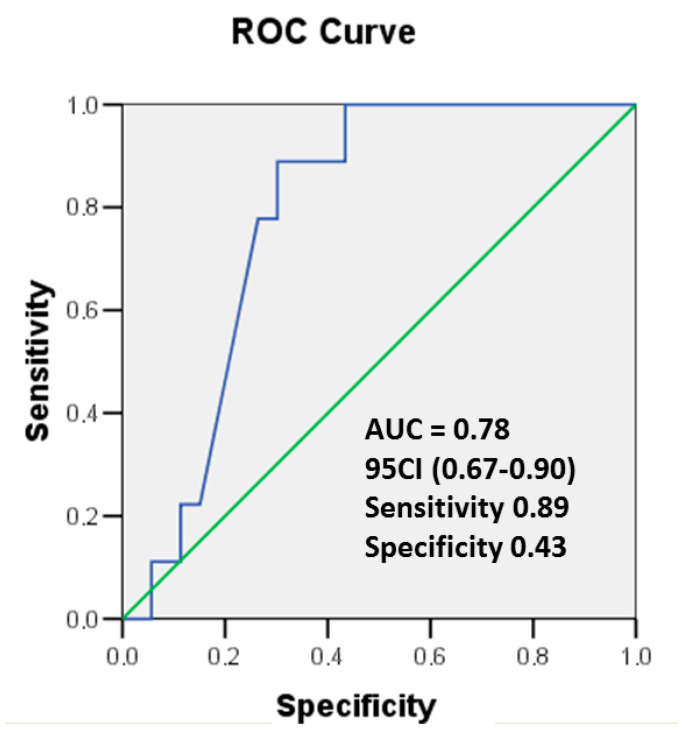
ROC curves showing the predictive value of Δ hsTnT on the drop in LV EF function ≥ 10% or ≤50%. Area under the curve (AUC) as indicated. hsTnT, high-sensitivity troponin T.

**Figure 5 cancers-17-02412-f005:**
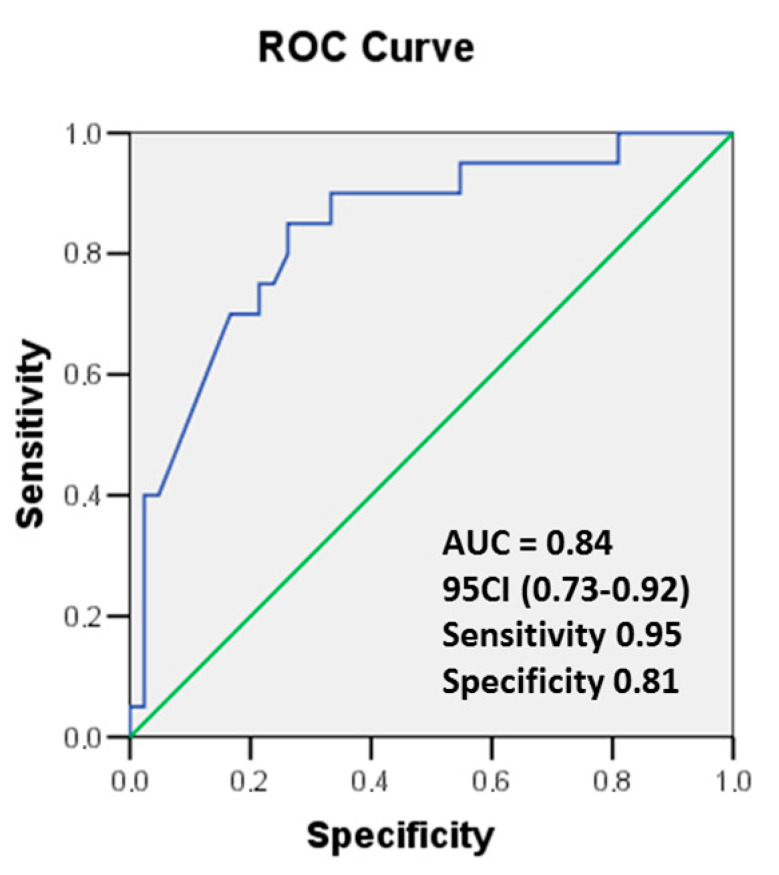
ROC curves showing the predictive value of Δ hsTnT on the drop in LV EF function ≥5% or <10%). Area under the curve (AUC) as indicated. hsTnT, high-sensitivity troponin T.

**Table 1 cancers-17-02412-t001:** Baseline characteristics of BC patients included in the study (M ± m).

Variable	Value (*n* = 60)
**History and Demography**	
Sex (female), *n* (%)	60 (100)
Age, years	48.6 ± 1.3
Patients of age > 65 years, *n* (%)	5 (7)
Smoker, *n* (%)	2 (3)
BMI, kg/m^2^	27.8 ± 0.8
Obesity, *n* (%)	12 (16)
Dyslipidemia, *n* (%)	21 (27)
**Hemodynamic Measurements**	
Systolic BP, mm Hg	128.2 ± 3.8
Diastolic BP, mm Hg	82.9 ± 1.8
Heart rate, b.p.m.	84.1 ± 2.5
LV EF, %	61.6 ± 0.5
**Comorbidities**	
Coronary Artery Disease, *n* (%)	3 (4)
Diabetes Mellitus, *n* (%)	2 (3)
Hypertension, *n* (%)	23 (30)
**Biomarkers**	
hsTnT level in plasma, ng/L	5.5 ± 1.4

Data are shown as *n* (%); BMI, body mass index; BP, blood pressure; LV EF, left ventricular ejection fraction.

**Table 2 cancers-17-02412-t002:** Cancer stage and cancer treatment.

Variable	Value (*n* = 60)
**Cancer Stage**	
I	2 (3)
II	18 (30)
III	35 (59)
IV	5 (8)
**Cancer Treatment**	
Anthracyclines, *n* (%)	60 (79)
Cumulative antracycline dose, (mg/m^2^)	258.6 ± 13.2
Trastuzumab, *n* (%)	30 (39)
Radiation therapy, *n* (%)	25 (32)

Data are shown as *n* (%).

## Data Availability

Research data are available on request.

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
