# Peer review of "The Prognostic Value of High-Sensitive Troponin T Rise Within the Upper Reference Limit in Breast Cancer: A Prospective Pilot Study"

_cancers, 2025, doi:10.3390/cancers17142412_

Round 1
Reviewer 1 Report
Comments and Suggestions for Authors
The authors attempt to investigate the role of high-sensitive cardiac troponin T increase below the upper limit of normal in patients with breast cancer. 60 women were finally included in the analysis which reported also echocardiogram data. Final results indicated that Δ hsTnT had a good predictive value for LV dysfunction (LVEF drop >10%). The increase of hsТnТ >81% was determined as the optimal threshold value for detecting early biochemical cardiotoxicity.
In my opinion, the paper is overall well-written and clear in its content. Statistica analysis appears to me simple but appropriate for the aim of the paper. However, I found significant limitations which would limit results apllicability in clinical practice and therefore publication.
-The authors state the most significant limitation is related to the very small sample size, which strongly prevents interpretation of results. As well, the inclusion of BC patients alone may represent a limitation as well.
-I could not find information of cancer stage at diagnosis, HER2 inhibitors administration lenght and pre-existing cardiovascular medications before enrollment. This could have influenced troponin behaviour. As well, I would ask the authors what medications were used in case of troponin rise and/or LVEF drop.
-Central illustration is missing.
Author Response
Comment 1: The authors state the most significant limitation is related to the very small sample size, which strongly prevents interpretation of results. As well, the inclusion of BC patients alone may represent a limitation as well.
Response 1: Thank you for pointing this out. We agree with this comment. Therefore, in the original version of the article, in the section on research limitations, we noted: 1) the small sample size, 2) the fact that this is a pilot study. Why we chose only patients with BC - because it is the best model for studying cardiotoxicity, given that this population has the highest incidence of cardiotoxicity (especially when combining anthracyclines and targeted therapy). We also mentioned this in the study limitations.
Comment 2: I could not find information of cancer stage at diagnosis, HER2 inhibitors administration lenght and pre-existing cardiovascular medications before enrollment. This could have influenced troponin behaviour. As well, I would ask the authors what medications were used in case of troponin rise and/or LVEF drop.
Response 2: Thank you for this comment. We have added information about the cancer stage at diagnosis, in the table.
During 6 month BC patients received from 2 to 7 cycles of HER2 inhibitors according to their hormone status and treatment protocol.
Before enrollment pts received cardiovascular medications:
- BB - n=4
- Indapamide - 2
- ACE inhibitors/ARBs= 6
- CCB = 4
- Metformine -1
- Statins -6
In case of troponin elevation more than normal ranges and/or LVEF drop BB, ACEI, MRA, loop diuretics etc. or their combination had been prescribed in accordance with ESC Cardio-Oncology recommendations 2022.
Comment 3: Central illustration is missing.
Response 3: Thank you for this comment. We prepared a central illustration and uploaded it to the article initially. Let's add the central illustration again.
Reviewer 2 Report
Comments and Suggestions for Authors
In this study, the authors would like to explore how troponin T levels will change upon chemotherapy treatment. Their study includes 60 patients with breast cancer and followed was supposed to be at 3 and 6 months post-chemo. I think studies such as this one are needed and relevant. But, it is underpowered and several pieces of information missing. I have outlined below what is needed:
- Troponin levels during cardiac arrest need to be stated to compare to what changes they are observing;
- Instead of stating TnT levels change by > 81%, it is best to state both the change % and actual number;
- "5 thousand" on line 49 needs to be numerically stated;
- Why did they not look at both cTnI and cTnT or BNP levels? What is the literature evidence for looking at only cTnT?
- Why do they think the excluded patients with > 14 ng/L cTnT had those levels? What were the actual cTnT levels of these patients?
- WHat is the % of breast cancer patients that have high baseline cTnT levels? And why?
- The excluded patients are interesting to look at - why did they not examine their cardiac function by EKG?
- In Table 1, the number of patients is only 57...what happened to the other 3 as the total was supposed to be 60?
- What was LV EF at > 150% rise in cTnT? How did those patients feel?
- They need to graphically stratify patients that had 10-305% increase in TnT as that is a large data spread. This will help to really understand how patients need to be treated if they have 10% or 305% increase in cTnT. In addition, it will help the reader understand the power of the study.
- Figure 2 needs to show all data points under the bar plot especially since p is close to 0.05.
- Why was data not shown for 3 months but only 6 months?
- Please explain ROC in more details so we can understand Fig 3 and 4. Were these figures at 6 months? What do they really mean?
- What does the "1" on the x-axis of figure 3 mean?
- Overall, more explanation of results is needed.
If these points can be addressed, I think the study will be greatly improved.
Author Response
Comment 1: Troponin levels during cardiac arrest need to be stated to compare to what changes they are observing.
Response 1: Thank you for pointing this out. We agree with this comment. It is important to evaluate this indicator over time. Troponin levels were studied before the start of cancer treatment and during 6 months
Comment 2: Instead of stating TnT levels change by > 81%, it is best to state both the change % and actual number.
Response 2: Thank you for this comment. Since the initial troponin level is different for all patients, we believe that displaying the changes in percentage (%) is more informative and visual. We added Figure 2 with distribution of hsTnT changes during 6 Months.
Comment 3: "5 thousand" on line 49 needs to be numerically stated.
Response 3: Thank you for pointing this out. We agree with this comment.
Comment 4: Why did they not look at both cTnI and cTnT or BNP levels? What is the literature evidence for looking at only cTnT?
Response 4: Thank you for this comment. cTnT assays are standardized because all commercial cTnT assays are licensed from Roche. In contrast, cTnI assays vary widely across platforms (Abbott, Siemens, Beckman, etc.) inter-assay variability makes multicenter comparison harder. Therefore, for clinical trials or multi-center registries, cTnT offers greater consistency. Many early cardio-oncology studies (especially with anthracyclines, trastuzumab, and TKIs) used hs-cTnT to monitor subclinical myocardial injury. For example: Cardinale et al., JACC 2004, 2006, 2010: pivotal studies using cTnT as a predictor of later LVEF decline. JACC CardioOncology guidelines recommend either hs-cTnT or hs-cTnI, but much early data is from hs-cTnT. We did not use BNP because they do not reflect myocardial damage.
Comment 5: Why do they think the excluded patients with > 14 ng/L cTnT had those levels? What were the actual cTnT levels of these patients?
Response 5: Thank you for this comment. Excluding patients with baseline hsTnT >14 ng/L (i.e., above the 99th percentile upper reference limit for high-sensitivity troponin T) is a common practice in cardio-oncology and cardiac biomarker studies to ensure that elevations represent new, treatment-related myocardial injury, not pre-existing cardiac conditions. We hypothesized that the rise of hsTnT even within ULN has important clinical significance for CV outcomes.
Comment 6: What is the % of breast cancer patients that have high baseline hhsTnT levels? And why?
Response 6: Thank you for pointing this out. Among 88 subjects, 4 breast cancer patients had high baseline hscTnT levels (indicated under Figure 1). Patients excluded due to cTnT >14 ng/L likely had pre-existing cardiac stress or injury (e.g., CAD, CKD, advanced cancer), and we excluded them to ensure a cleaner assessment of de novo therapy-related myocardial injury.
Comment 7: The excluded patients are interesting to look at - why did they not examine their cardiac function by EKG?
Response 6: Thank you for this important question. Excluded patients underwent cardiac function monitoring using echocardiography, EKG, and biomarkers in accordance with current recommendations. However, since the initial increase in troponin was an exclusion criterion, such patients were not included in the study.
Comment 8: In Table 1, the number of patients is only 57...what happened to the other 3 as the total was supposed to be 60?
Response 8: Thank you for this comment. This is technical mistake. Really in analysis we included 60 patients.
Comment 9: What was LV EF at > 150% rise in hscTnT? How did those patients feel?
Response 9: Thank you for pointing this out. The main objective of the study was to investigate whether troponin elevation within the upper limit of normal had clinical significance. Patients with >150% increase in hscTnT had a significantly higher rate of cardiotoxicity.
Comment 10: They need to graphically stratify patients that had 10-305% increase in TnT as that is a large data spread. This will help to really understand how patients need to be treated if they have 10% or 305% increase in hscTnT. In addition, it will help the reader understand the power of the study
Response 10: We prepared and added Figure 2 with distribution of hsTnT changes during 6 Months.
Comment 11: Figure 2 needs to show all data points under the bar plot especially since p is close to 0.05.
Response 11: Thank you for this comment. We have used a vertical bar plot to represent a comparative statistically significant results in EF between baseline and 6 months.
Comment 12: Why was data not shown for 3 months but only 6 months?
Response 12: Thank you for this very insightful question. If a study on cardiac damage during BC chemotherapy only reports 6-month data and not 3-month data, there are a few likely scientific and methodological reasons - especially if the study tracked troponin T and LVEF as markers of cardiotoxicity. Early subclinical injury (e.g., troponin rise) may not cause measurable LVEF decline at 3 months. Many studies have shown that clinically meaningful LV dysfunction typically emerges at 4-6 months or later. Guidelines typically recommend cardiac imaging every 3-6 months during HER2 therapy, so 6-months aligns with real-world practice.
Comment 13: Please explain ROC in more details so we can understand Fig 3 and 4. Were these figures at 6 months? What do they really mean?
Response 13: Thank you very much for this suggestion. We have added an explanation of ROC in more detail. Yes, these figures are at 6 months. We're referring to a ROC curve evaluating whether Δ hsTnT (change in high-sensitivity cardiac troponin T) predicts a ≥10% drop in LVEF or an absolute LVEF ≤50% - both of which are clinically relevant markers of cancer therapy-related cardiac dysfunction. AUC = 0.78: This suggests that Δ hsTnT is a moderately good predictor of LVEF drop ≥10% or LVEF ≤50%.
Comment 14: What does the "1" on the x-axis of figure 3 mean?
Response 14: Thank you for this comment. We delete the "1" from the x-axis of figure 3 mean.
Comment 15: Overall, more explanation of results is needed.
Response 15: Yes, we agree. Further prospective, randomized, multicenter trials are needed to confirm our findings in BC and other cancer sites.
Round 2
Reviewer 1 Report
Comments and Suggestions for Authors
Data on GLS and diastolic function seem to be missing. These missing data should be clearly stated in the limitation section considering that the available definition of CTRCD encompasses also GLS. Moreover, knowledge is rising on the role of diastolic dysfunction as early marker of cardiotoxicity with tight correlation with cardiac biomarkers. Please consider citing these recent acquisitions: PMID: 40335213; PMID: 39025347; PMID: 39479333.
Author Response
Comment 1: Data on GLS and diastolic function seem to be missing. These missing data should be clearly stated in the limitations section, considering that the available definition of CTRCD also encompasses GLS.
Response 1: Thank you for bringing this to our attention. We agree with this comment. We updated the limitation section.
Comment 2: Moreover, knowledge is rising on the role of diastolic dysfunction as early marker of cardiotoxicity with tight correlation with cardiac biomarkers. Please consider citing these recent acquisitions: PMID: 40335213; PMID: 39025347; PMID: 39479333.
Response 2: Thank you for this comment. We have added two recent publications.
Barac A, Brazile TL. Did You Check Diastolic Function During Cancer Treatment?: It's Not Just Systole's Show. JACC Cardiovasc Imaging. 2025 May;18(5):554-556. doi: 10.1016/j.jcmg.2025.02.006. PMID: 40335213.
Camilli M, Cipolla CM, Dent S, Minotti G, Cardinale DM. Anthracycline Cardiotoxicity in Adult Cancer Patients: JACC: CardioOncology State-of-the-Art Review. JACC CardioOncol. 2024 Sep 17;6(5):655-677. doi: 10.1016/j.jaccao.2024.07.016. PMID: 39479333; PMCID: PMC11520218.
Round 3
Reviewer 1 Report
Comments and Suggestions for Authors
The authors replied to my comments.